

# NN4CAST: An end-to-end deep learning application for seasonal climate forecasts

Víctor Galván Fraile[1], Belén Rodríguez-Fonseca[1,2], Irene Polo[1], Marta Martín-Rey[1], and María N. Moreno-García[3]

[1]Group on Tropical Climate Variability and Atmospheric Teleconnections (TROPA), University Complutense of Madrid (UCM), Madrid, Spain
[2]Institute of Geosciences (CSIC), Madrid, Spain
[3]Data Mining Research Group, University of Salamanca (USAL), Salamanca, Spain

**Correspondence:** Víctor Galván Fraile (vgalva01@ucm.es); Belén Rodríguez-Fonseca (brfonsec@ucm.es).

**Abstract.** Predicting climate variables at seasonal time scales is crucial for climate services. At these time scales, the most important driver of the atmospheric variability patterns is the ocean, which can trigger local perturbations that could affect the climate of remote areas through different teleconnection mechanisms. Dynamical models are not always effective in overcoming the challenges posed by complex climate processes and may show a low signal-to-noise ratio in the response of some phenomena (such as ENSO). Recently, statistical approaches, which focus on the relationships between predictor and predictand fields, have gained popularity due to advances in computing capabilities and the availability of extensive meteorological datasets. This is particularly evidenced in the development of short-range weather prediction data-driven models. However, seasonal prediction remains challenging, especially in regions where there are complex non-linear remote interactions of non-stationary and seasonal dependent signals from different sources of predictability of the Earth system. For this reason, we have developed a non-linear modelling tool, named as Neural Network foreCAST (NN4CAST). It is a deep learning model library which provides a versatile tool for operational prediction and non-linear statistical analysis, which can enhance seasonal forecast accuracy and understanding of the underlying dynamics. It is demonstrated how NN4CAST is able to provide accurate seasonal predictions in regions where the atmospheric response to the ocean is mostly linear (i.e., tropics) as well as in remote areas through atmospheric teleconnections, where the signal is highly non-linear (i.e., North Atlantic). It is in those remote regions where the dynamical models fail to capture the climate signal, and where the deep learning models may be more useful for a seasonal data-driven or hybrid prediction system. Therefore, the application of this model could provide accurate and reliable forecasts that can be useful for the different strategic societal sectors such as marine ecosystems, health or energy, which require predictions on these time scales.

## 1 Introduction

Climate prediction is the field of study that aims to forecast the future state of the Earth's climate system. The predictions can span from subseasonal and seasonal to decades or centuries. Specifically, seasonal forecasting attempts to provide useful information about the climate that can be expected in the coming months. It is crucial for different sectors such as agriculture,





water resource management or disaster preparedness. Seasonal predictions are mainly justified by the existence of interactions between the atmosphere and the slow, and predictable, variations in some of the components of the climate system, such as soil moisture, snow cover, stratospheric circulation or the ocean Sea Surface Temperatures (SSTs). Specifically, tropical SSTs are demonstrated to be one of the most important sources of predictability at seasonal time scales (Shuila and Kinter III (2006); Kirtman and Pirani (2009); Ineson and Scaife (2009)). Earth system models (ESMs) simulate the climate system by solving the mathematical equations that represent the interactions between the components of the climate system. However, they have errors due to not only the numerical resolution of the system of equations, but also to the uncertainty in the initial and boundary conditions. When it comes to seasonal time scales, different challenges arises:

1. Seasonal predictions rely on the correct representation of not only local processes such as deep convection, but large scale ones as global teleconnections. A missrepresentation of either of them leads to poor performances in certain remote regions, where the interaction of the different signals is non-linear and seasonally dependent (Gleckler et al. (2008); Doblas-Reyes et al. (2013)).

2. Multidecadal ocean variability and the Global Warming trend alters the global circulation and, thus, the way in which atmospheric teleconnections (i.e., Rossby waves) propagate, introducing non-stationarities in the system (López-Parages et al. (2015); Weisheimer et al. (2017)).

3. Oceanic patterns of variability, which may provide seasonal predictability in certain regions such as the Atlantic Niño or the North-Atlantic SSTs, are not well represented by current models (Richter and Tokinaga (2020); Roberts et al. (2021)).

4. The generation of coupled simulations at these timescales demands substantial number of processors which, in turn, constrains the potential for enhancement in spatial resolution.

Alternatively to ESMs, statistical methods directly focus on the patterns and relationships between different climate variables with different time lags. Another important advantage is their high computational efficiency, as they only require significant computing power during the training phase. However, they could struggle to capture complex non-linear relationships and non-stationarities in the modelled relationships.

Artificial Intelligence (AI) is a broad area of computer science research, primarily concerned with the formulation and implementation of methodologies and software frameworks that enable machines to perceive environmental cues. These methods use principles of learning and intelligent decision making to optimise the achievement of predetermined goals (Russell and Norvig (2016)). Machine Learning (ML) is a subfield of AI that focuses on the development and study of statistical algorithms. Unlike traditional data analysis methods that rely on fitting data to a specific function, ML algorithms improve handling of complex, high-dimensional data that cannot be adequately represented by a predefined mathematical model. These methods can learn from data, generalise to unseen data, and perform tasks without explicit instructions (Samuel (1959)). Deep learning (DL) is a subfield of ML techniques which have emerged as an evolution of neural networks by incorporating multiple hidden layers to enhance the automatic learning and extraction of complex patterns from data. This category includes Convolutional



Neural Networks (CNNs) whose main characteristic is their ability to automatically and adaptively learn spatial hierarchies of features through convolutional layers, which reduces the need for manual feature extraction. CNNs provide a more powerful and scalable approach to learning from complex data structures compared to classical techniques (LeCun et al. (2015)). This leads to superior performance in tasks such as image recognition, object detection, and medical image analysis, surpassing

traditional methods in accuracy and efficiency (Rawat and Wang (2017)).

Deep Learning is an increasing powerful and popular tool, which has proven to be computationally efficient, as well as being a way to produce accurate weather and climate predictions (Rew et al. (2006)). The potential of DL in this field is evidenced by the development of different data-driven models not only from meteorological agencies and researchers, but also from private companies. It is important to clarify that data-driven models are constructed based on empirical data, rather than through

explicit theoretical equations like dynamical models. Some examples of these data-driven models are: PanguWeather, which is a 3D Earth-specific transformer module created by the Huawei Cloud group (Bi et al. (2022)); GraphCast, developed by researchers from DeepMind and Google (Lam et al. (2022)). These are two examples of weather data-driven models, but there are a wide range of models with different architectures. The application of ML techniques for climate modelling is not new at all. Several techniques, including linear regressions, principal component analysis, correlations, and maximum covariance

analysis, have been widely applied (Wilks (2014); Suárez-Moreno and Rodríguez-Fonseca (2015); Rieger et al. (2021)). The main advantage of these models are they simplicity and the white box structure which allows the user to understand the model behaviour on the predictions. The common idea behind all of them is the assumption of linear relationships between the predictor and predictand fields. This is critical when modelling the Earth system, which is mainly composed by complex non-linear components, leading to poor performances in certain regions. There are, however, other classes of ML which are much

less widely used within the climate modelling (i.e., neural networks), specifically in the sub-seasonal to seasonal time-ranges (Vitart and Robertson (2018)). These DL models have the potential to address the current limitations of seasonal prediction models, not only by enhancing prediction accuracy but also by investigating the underlying attribution mechanisms. Some studies have been conducted in this direction, for instance, to enhance understanding of the dynamics of El Niño-Southern Oscillation ( Ham et al. (2019); Shin et al. (2022)) or the teleconnection associated with the Madden-Julian Oscillation (Mayer

and Barnes (2021)). Nevertheless, there is currently no tool that enables such techniques to be deployed in a systematic manner.

For these reasons, we have created the Neural Network foreCAST (NN4CAST) application, which allows the creation of deep learning models that allows for non-linear modelling of climate teleconnections. Furthermore, one of the main objectives is to avoid treating it as a black box and to understand the origin of the predictability and its sensitivity to the change of the training period. This final factor is an important added value of NN4CAST models compared to other seasonal forecasting

models. It enables the analysis of predictability, as well as the examination of teleconnections, their modulations, the identification of windows of opportunity and the production of attributions of the predictions over certain target regions. NN4CAST is implemented as a Python library designed for use in applications with small or large datasets for subseasonal, seasonal or decadal predictions. This tool could be coupled to more complex frameworks, such as ESMValTool, a community tool of diagnostic and performance metrics for evaluating Earth system models (Righi et al. (2020)). Scientific tools are commonly

written using low-level compiled languages such as C or C++, due to their greater computational efficiency compared to high-



level interpreted languages such as Python. However, Python has become the dominant programming language for ML tools, including the libraries used in NN4CAST. This highlights the importance of NN4CAST not only for research purposes but also for operational seasonal forecasting applications.

This paper introduces the NN4CAST package, first by briefly describing the theoretical background of neural networks and explaining each of its features. Secondly, it demonstrates the application to a specific example, highlighting the library's usefulness and limitations in this case. The code is available on GitHub and in a Zenodo folder Galván Fraile et al. (2024). The paper is structured as follows: Section 2 explains fundamental concepts of deep learning, and section 3 describes the NN4CAST methodology and implementation. In section 4, an example of an NN4CAST application is presented for predicting the mean sea-level pressure (SLP) anomalies over the whole Northern Hemisphere (NH) on early-winter (November-December, ND) using the information from the SSTs from the Pacific Ocean in the previous autumn (September-October, SO). Finally, Section 5 describes the main conclusions and future lines of research.

## 2 Theoretical framework

In a seasonal prediction system, a predictor field (X) is initialized in a particular time in order to forecast the predictand field (Y). The predictor exhibits inertia, which allows for a delayed forecast of the predictand. There is a wide variety of statistical seasonal methods, but in this case a deep learning approach will be used.

Deep learning models represent an advancement in the field of artificial neural networks, comprising multiple hidden layers that enable the model to more effectively capture the non-linear behaviour of complex systems. In a neural network (NN) approach, in addition to X and Y, a number of hyperparameters must be defined. Recall that the hyperparameters are the configuration settings for a ML algorithm that are set before the learning process begins, while the parameters of the model are those learned during the training phase. It is important to note that the performance of NN based models strongly depends not only on the election of the different hyperparameters (number of layers, number of neurons per layer, activation functions, optimization algorithm, etc), but on the quality and size of the dataset as well. The basic unit, each neuron, has $\mathbf{x} \in \mathbb{R}^n$ input values and $\mathbf{w} \in \mathbb{R}^n$ weights. The output of each individual neuron ($j$) is determined by the sum of the products between each neuron ($i$) in the preceding layer and their respective weights, aggregated across all ($n$) neurons in the previous layer, as expressed in the following formula:

$$\mathbf{z_j} = \mathbf{f}\left(\sum_{i=1}^{n} \mathbf{w_i}\mathbf{x_i} + \mathbf{b}\right) \tag{1}$$

being $\mathbf{b} \in \mathbb{R}$ the bias value and $\mathbf{f}$ a non-linear activation function (Géron (2022)). The activation function is applied to the output of a neuron in a neural network, which could introduce non-linearity into the network, allowing it to learn and represent complex patterns in the data. The output from each neuron ($\mathbf{z_j}$) is propagated along the different neurons grouped by the layers of the model, reaching by this way the prediction from the neural network model ($\hat{\mathbf{y}}$).

Convolutional Neural Networks are multi-layer NNs with, at least, one convolutional layer. The key idea behind this type of layers is that each neuron is not connected to every single neuron in the previous layer, but only to those in the receptive field



(Géron (2022)). The convolution operation involves sliding a filter (kernel) over the input field and performing element-wise multiplication and summation. By this way, the network can focus on low-level features at each layer, taking into account the
spatial relationships between separate features.

Once the model has been constructed, it must be trained. The most common approach to this is to divide the dataset into three distinct subsets: training, which is the one used to tweak to model parameters; validation, used for evaluate the model performance in each training iteration; and testing, used to evaluate the final performance of the model in an independent dataset. To evaluate the predictions, a loss function is defined, which quantifies the difference between the model predictions
and the actual observations. The model tries to minimize this function over the training set by updating the weights of the different layers. The most common loss function for regression tasks is the Mean Squared Error (MSE), which is given by

$$MSE = \frac{1}{m} \sum_{i=1}^{m} \left( y^{(i)} - \hat{\mathbf{y}}^{(i)} \right)^2 \tag{2}$$

where $m$ is the number of instances in the dataset, $\hat{\mathbf{y}}^{(i)}$ is the predicted value of the predictand variable for the $i^{th}$ instance in the dataset and $y^{(i)}$ is its corresponding real value (i.e., the ground truth) (Wilks (2008)). However, it is possible to modify
this objective function to any other differentiable loss function. Physically constrained loss functions can also be added to benefit the model during both the training and testing phases. The main problem when training these models is the so called bias-variance trade-off, which arises from the need to balance two sources of error in predictive models:

· Bias (Underfitting): This error occurs when a model is too simple and fails to capture the underlying patterns in the data. A high bias model typically leads to underfitting, where the model is unable to learn complex relationships.

· Variance (Overfitting): This error occurs when a model is too complex and learns the noise and fluctuations in the training data rather than the underlying patterns. A high variance model often performs well on training data but poorly on unseen data due to its inability to generalize.

The optimal model would be one that does not suffer too much from both. There are two common techniques that help to address this trade-off. On the one hand, regularization, which introduces a penalty term to the loss function during training,
reducing the complexity of the model and avoiding overfitting. There are different types of regularization, namely, Lasso (L1), Ridge (L2) and a combination of them (L1_L2). The main difference between L1 and L2 is that the penalty is proportional to the absolute values and the squared values of the model coefficients, respectively. On the other hand, the dropout technique, in which, during training, randomly selected neurons are temporally removed ("dropped out") with a specified probability (Géron (2022)).

Once the model has been trained, the parameters are fixed and used to make the predictions over the testing dataset. To evaluate its performance, different metrics could be used, such as the Root Mean Squared Error (RMSE), which is the square root of the MSE, or the Anomaly Correlation Coefficient (ACC), which is given by

$$ACC = \frac{\sum_{i=1}^{m} y^{(i)} \cdot \hat{\mathbf{y}}^{(i)}}{\sqrt{\sum_{i=1}^{m} \left( y^{(i)} \cdot \hat{\mathbf{y}}^{(i)} \right)^2}} \tag{3}$$



where the symbology is the same as for the previous metric (Wilks (2008)). Furthermore, the best way to evaluate a model is

through the process of cross-validation, in which it takes a number of k-folds to train and evaluate iteratively the model on the whole dataset. This allows to maximize the use of available data for both, training and testing. This methodology allows to produce a hindcast for the entire dataset period, enabling the possibility of applying a leave-one-out cross-validation to perform a more detailed hindcast.

In order to facilitate a more comprehensive analysis of the underlying mechanisms driving the predictions, the relative
importance of a specific region of the predictor field, can be identified in the field of the Explainable AI (XAI). XAI aims to enhance the added value of artificial intelligence methods by enabling the identification of the most influential predictors in the prediction process. There are two main categories of methods: sensitivity methods, which assess the sensitivity of the output value to a specific predictor, and attribution methods, which determine the relative contribution of each predictor to the predictand (Guidotti et al. (2018)). One of the most attribution method is Integrated Gradients (Sundararajan et al. (2017)),
which addresses the issue of non-linear problems, where the derivative is not constant. It considers a reference (baseline) vector $\hat{\mathbf{x}}$ for which the model output is zero: $\hat{\mathbf{F}}(\hat{\mathbf{x}}) = 0$. The importance is computed as the product of the distance between the input within the reference point and the average of the gradients at points along the straight-line path from the reference point to the input feature. Specifically, the mathematical expression is given by

$$R_{i,n} = (x_{i,n} - \hat{\mathbf{x}_i}) \frac{1}{m} \sum_{j=1}^{m} \frac{\partial \hat{\mathbf{F}}}{\partial X_i} \bigg|_{X_i = \hat{\mathbf{x}_i} + \frac{j}{m}(x_{i,n} - \hat{\mathbf{x}_i})} \tag{4}$$

where $X_i$ are the input features, $R_{i,n}$ the relevance of feature at grid point $(i)$ for the model prediction of sample $(n)$, $\hat{\mathbf{F}}$ the function learned by the model and $(m)$ the number of steps in the Riemann approximation (Mamalakis et al. (2022)).

## 3  NN4CAST methodology & implementation

In the previous section, the most important aspects of DL have been delineated, including recent advancements such as XAI. Based on these, we have constructed the NN4CAST library, which enables the integration of these methodologies within a
unified framework, through the following steps:

- · Preprocessing of the datasets according to user targets, including: reading the data, trimming the region of interest, computing anomalies and trend elimination.

- · Creation of a deep neural network to generate a seasonal prediction model, adjusting the weights by the minimization of a loss function (MSE).

- · Application of regularization methods to avoid overfitting problems.

- · Evaluation of model performance via cross-validation with different metrics (ACC and RMSE).

- · Analysis of attributions in the predictor to explain predictions about a target region in the XAI framework.




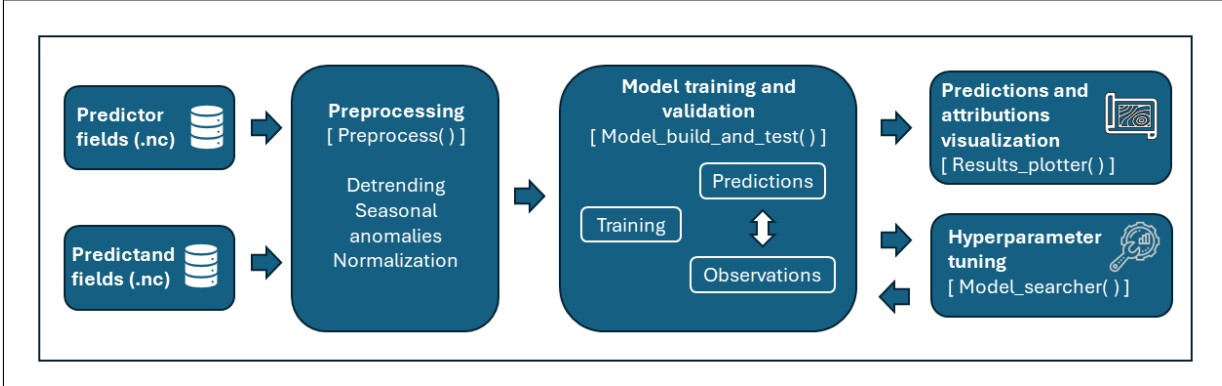

**Figure 1.** Flowchart describing the methodology and application setup using the NN4CAST library.

The whole procedural workflow is depicted in Fig. 1. The application begins with the loading of the predictor and predictand datasets, as well as the definition of the hyperparameters of the model. Then, a preprocessing of them is done by applying the `Preprocess()` function. After that, the model is built, trained and tested in the next step by calling the `Model_build_and_test()` functionality. Finally, the individual predictions and attributions can be seen with the `Results_plotter()` functionality. Optionally, the `Model_searcher()` function enables the possibility of searching for a more optimal model by tuning its hyperparameters. The complete configuration can be accomplished using the primary software package called `nn4cast.predefined_classes`.

The main peculiarity of NN4CAST is that it has been designed to be used in the field of seasonal prediction, trying to facilitate experiments that can help to cope with typical problems of seasonal forecasting. Concretely:

1. the problem of representation of teleconnections. The models created by NN4CAST can be run for different regions of the predictor field, in order to be able to identify the relative contribution of each of them.

2. the non-stationarity problem. The models can be run for different training periods in order to produce predictions in a particular period. This allows to analyze the non-stationarity behaviour of the relationships.

3. the computational resources problem. The models are capable of performing simulations within a few minutes on a standard computer.

4. the black box nature of DL. The models have the advantage of providing insightful attributions of the predictions which allow to identify the underlying physical mechanisms.

## 3.1 Configuration of hyperparameters and Preprocess

NN4CAST employs datasets stored in the netCDF4 format (Rew et al. (2006)), a widely adopted standard within the realm of Earth data science. These datasets are structured as space-time matrices and require three coordinates: time, latitude, and





longitude. Additionally, NN4CAST employs a dictionary to define the hyperparameters of the model. Listing 1 demonstrates how to create this dictionary and save it as a .yaml file. YAML is a text-based file format suitable for storing structured data, such as model configurations. Saving the dictionary as a .yaml file enables the documentation of the experiment details for easy retrieval and sharing. The hyperparameters required by the application are described in Table 1, giving a brief description of their functionality.

**Table 1.** Name and explanation of the hyperparameters required by the application.

| Hyperparameter name | Description |
|---|---|
| $path\_x$, $path\_y$ | Path to the X (predictor) and Y (predictand) datasets files. |
| $time\_limits$ | Define the time limits (in years) to select from the original datasets. They should be in the format ['yyyy', 'yyyy'], where the first year is the start and the second year is the end of the period, separated by commas. For example, ['1975','2015']. |
| $jump\_year$ | Define if the X and Y fields are from different years. If 0 means that they come from the same year, and number indicate the lead time between them (default=0). For example, if X is from September and Y is from February of the following year, then $jump\_year = 1$. |
| $months\_x$, $months\_y$ | Select the months of X and Y, knowing that January=1 and December=12. |
| $months\_skip\_x$, $months\_skip\_y$ | Define the months to skip. This is necessary if if several months are considered to analyse seasons involving changes of the year. For example, if X samples are from Dec-Jan-Feb and the data spans from 1950 to 2019, it is necessary to set $months\_skip\_x$ = ['1950-01', '1950-02', '2019-12'], to delete the Jan-Feb from the first year and the Dec from the last one, in order to create seasonal means that makes climatic sense. |
| $train\_years$, $validation\_years$, $test\_years$ | Define the periods (initial year, final year) for training, validating and testing the model based on X years. |
| $lat\_lims\_x$, $lon\_lims\_x$, $lat\_lims\_y$, $lon\_lims\_y$ | Define the latitude-longitude regions for the predictor (X) and predictand (Y) fields. For the latitudes, it can be selected in any order (smaller first or last), whereas, for the longitudes, it is needed to put the smaller first, and either in the format: -180-(+180) or 0-360. |
| $name\_x$, $name\_y$ | Specify the variable names, as defined in the original netcdf files. |
| $mean\_seasonal\_method\_x$, $mean\_seasonal\_method\_y$ | Define if computing the seasonal means (=True) or aggregates (=False). |
| $scale\_x$, $scale\_y$ | Define if scaling the data, if not, set as 1. For example, if the pressure data is given in Pa and you want to have the data in hPa, then, a scale of 100 need to be applied. |



| | |
|---|---|
| $regrid\_degree\_x$, $regrid\_degree\_y$ | Define the interpolation degree for regridding the data. Values greater than 0 apply regridding, while 0 means no regridding (default=0). |
| $detrend\_x$, $detrend\_y$ | Define if performing a backward moving average detrend of the data. If True, it is needed to define the window size to compute the detrend iteratively. |
| $detrend\_x\_window$, $detrend\_y\_window$ | Define the size of the past window to compute the averages to do the detrend (default=10). |
| $layer\_sizes$ | Define the number of neurons per layer of the model as a list (default=[1024, 256, 64]). |
| $activations$ | Define the activation function of each layer of the model as a list (default= [tf.keras.activations.elu, tf.keras.activations.elu, tf.keras.activations.elu]). |
| $kernel\_regularizer$ | Define whether using or not a regularizer. The options are: $l\_1$, $l\_2$, $l1\_l2$ or $None$ (default=['$l1\_l2$']). |
| $learning\_rate$ | Set the learning rate of model hyperparameters update (default=0.0001). |
| $epochs$ | Set the number of epochs to train the model, knowing that there is an early stopping when no increase is reached in the validation loss on the last 25 epochs (default=2500). |
| $num\_conv\_layers$ | Set the number of convolution layers to apply (default=0). If at least 1 convolution layer is applied, you can define also the $number\_filters$, $kernel\_size$ and $pooling\_size$. |
| $use\_batch\_norm$, $use\_initializer$ | Set to True to perform batch normalization and Kaiming He initialization (He et al. (2015)) of the parameters in each layer of the model. (default=False). |
| $use\_dropout$, $dropout\_rates$ | Set to True to perform dropout, giving also the dropout rates as a list. (default=False). |
| $use\_init\_skip\_connections$, $use\_inter\_skip\_connections$ | Set to True to use initial or intermediate skip connections as in the U-Net architecture (Ronneberger et al. (2015)) (default=False). |
| $units\_x$, $units\_y$ | Set the units of the input and output fields to fill the labels in the plot. |
| $region\_predictor$ | Set the name of the predictor region to be included as information for the figures. |
| $p\_value$ | Define the $p\_value$ used to establish the confidence interval through the T-test method for assessing statistically significant values of the model skill. |
| $outputs\_path$ | Select the directory where the plots and datasets will be saved. |

The next step is to preprocess the data of the predictor and predictand, in order to compute the operations given in the hyperparameters dictionary: regriding the dataset,seasonal mean anomalies, detrending, etc. It is important to note how the
detrending works. It is made by applying a backward moving average (BMA) algorithm, which computes the running mean




of the previous years using a sliding window and substracting it to the following value (Raffalovich (1994); Alvarez-Ramirez et al. (2005)). By this way it avoids to introduce future information in the preprocessing phase. For example, if the size of the sliding window is 10 years, the detrend will be applied from year 11 onwards. Notice that this detrending methodology is not a usual linear detrend, but that low frequencies may also be filtered out in the process, depending on the size of the sliding windows (larger windows will filter out only the lowest frequencies). The preprocess is done by applying the functionality `Preprocess()`, as shown in Listing 2, returning a dictionary with the datasets needed to train, validate and test the model.

### 3.2 Model Train-Validate-Test

The next step is to build up, train, validate and test the model within the function `Model_build_and_test_period()`. This is done as shown in Listing 3, where it returns a dictionary with two datasets: the predicted and observed fields on the test set. It also saves both datasets in the output directory defined in the hyperparameters dictionary, as well as some general plots with the metrics of model performance. Namely, the ACC and RMSE are calculated for both spatial (at each spatial point over time) and temporal (at each sample over space) dimensions. The ACC map provides information on regions of skill across the entire temporal period, while the ACC time series show how global skill varies yearly. The same interpretation applies to the RMSE maps. Additionally, if desired, the `Results_plotter()` function allows the possibility of plotting specific predictions against the observations in order to gain some intuition of the models performance, also plotting the predictor field, also shown in Listing 3. This is done by defining the specific years to plot as a list in the argument `years_to_plot`. If no years are selected, all the samples in the dataset will be plotted.

### 3.3 Model cross-validation & Explainable AI

Once the model has been trained, validated and tested on the selected periods, it is convenient to apply a cross-validation technique to obtain more objective performance metrics that are not so subject to the training-validation-testing periods chosen. Recall that a usual division of the dataset into this three subsets is 70%, 10% and 20%, respectively (Géron (2022)). With this in mind, the `Model_build_and_test()` function also allows the possibility of applying a cross-validation along the dataset, by defining the number of folds in which the dataset is divided (setting the command $n\_cv\_folds$ to the desired number of divisions). This is done by dividing the dataset into $k$ smaller sets. Then, for each $k$ fold, the model is trained in the $k-1$ other folds and tested on the isolated one, repeating this process iteratively through the k folds. Additionally, in order to prevent overfitting, a randomly selected 10 percent of each training set is reserved as a validation period. This allows for the implementation of an objective measure of when the model should stop the training phase. The maximum number of folds is the number of samples that the dataset contains, which will apply a leave-one-out cross-validation. This process, therefore, creates a hindcast where the model performance can be evaluated, which is shown in Listing 4.

As previously stated, a core objective of NN4CAST is to facilitate the generation of insightful attributions between predictions. To compute these attributions for a specific target region, first set the instance to `True` to enable the computation. Then, define the region as a list of latitude and longitude ranges (setting region_importances=[[latitude_min, latitude_max], [longitude_min, longitude_max]]), as shown in Listing 4. By this way, the model will apply the Integrated Gradients method-



```python
    from tensorflow.keras import activations
    import numpy as np
    from nn4cast.predefined_classes import Dictionary_saver

hyperparameters =
      'path_x' = "/path/to/your/data/HadISST1_sst_1870-2019.nc",
      'path_y' = "/path/to/your/data/slp_ERA5_1940-2023.nc",

      'time_limits' = [1940,2019],
'jump_year' = 0,

      'train_years' = [1940, 1989],
      'validation_years' = [1990, 1999],
      'testing_years' = [2000, 2019],

      'lat_lims_x' = [75, -20],
      'lon_lims_x' = [120, 280],
      'lat_lims_y' = [75, -20],
      'lon_lims_y' = [-180, +180],

      'name_x' = 'sst',
      'name_y' = 'msl',

      'months_x' = [9, 10],
'months_skip_x' = ['None'],
      'months_y' = [11, 12],
      'months_skip_y' = ['None'],

      'mean_seasonal_method_x' = True,
'mean_seasonal_method_y' = True,

      'regrid_degree_x' = 2,
      'regrid_degree_y' = 2,

'scale_x' = 1,
      'scale_y' = 100,

      'detrend_x' = True,
      'detrend_y' = True,
'detrend_x_window' = 50,
      'detrend_y_window' = 50,

      # Neural network hyperparameters (default ones).
      'layer_sizes' = [1024,256,64,256,1024],
'activations' = [activations.elu,activations.elu,activations.elu,
        activations.elu, activations.elu]
      'dropout_rates' = [0.1],
      'kernel_regularizer' = 'l2',
      'learning_rate' = 0.0001,
'epochs' = 2500,
      'num_conv_layers' = 0,
      'use_batch_norm' = True,
```





```
        'use_initializer' = True,
        'use_dropout' = True,
        'use_init_skip_connections' = False,
        'use_inter_skip_connections' = False,

57
        'units_x' = ['°C'],
        'units_y' = ['hPa'],
        'region_predictor' = 'Pacific',
        'p_value' = 0.1,
62      'outputs_path' = "/path/to/the/directory/Outputs_ND_sst_SO/"

    Dictionary_saver(hyperparameters)
```

**Listing 1.** Python example to create an instance of the dictionary of parameters and hyperparameters and how to save them using `Dictionary-saver` function.

```
    from nn4cast.predefined_classes import Preprocess

    dictionary_preprocess = Preprocess(dictionary_hyperparams=hyperparameters)
```

**Listing 2.** Python example to create an instance of `Preprocess` function to manipulate the predictor and predictand data and save them as a dictionary.

```
    from nn4cast.predefined_classes import Model_build_and_test

    outputs_hold_out = Model_build_and_test(dictionary_hyperparams=
      hyperparameters, dictionary_preprocess=dictionary_preprocess)
5
    Results_plotter(hyperparameters, dictionary_preprocess, rang_x=2.5, rang_y=10,
      predictions=outputs_cross_validation['predictions'],
      observations=outputs_cross_validation['observations'],
      years_to_plot=[2015], plot_with_contours=True)
10
```

**Listing 3.** Python example to create an instance of `Model-build-and-test` function to build the model using the hyperparameters selected. Then train, validate and test the model in the corresponding datasets. Also creating an instance of `Results-plotter` to plot some results and compare them with the observations.





ology included in the Alibi python library (Klaise et al. (2021)). This methodology benefits from the cross-validation approach

in order to obtain the attributions over the entire hindcast.

```
from nn4cast.predefined_classes import Model_build_and_test, Results_plotter

outputs_cross_validation = Model_build_and_test(dictionary_hyperparams =
    hyperparameters, dictionary_preprocess=dictionary_preprocess,
5   cross_validation=True, n_cv_folds=4, plot_differences=False, importances=True,
    region_importances=[[50,65],[-25,-10]])

Results_plotter(hyperparameters, dictionary_preprocess, rang_x=2.5, rang_y=10,
    predictions=outputs_cross_validation['predictions'],
10  observations=outputs_cross_validation['observations'],
    years_to_plot=[2015], plot_with_contours=True,
    importances=outputs_cross_validation['importances'],
    region_importances=outputs_cross_validation['region_attributed'])
```

**Listing 4.** Python example to create an instance of `Model-build-and-test` function to build the model using the hyperparameters selected. Then train, validate and test the model applying a cross-validation technique, with the number of folds defined. Also creating an instance of `Results-plotter` to plot some results and their attributions for the target region over the predictor field.

## 3.4 Model hyperaparameters tunning

NN4CAST also offers the possibility of optimising the hyperparameters of the model in order to achieve an overall better performance. First, a dictionary of the possible values of the hyperparameters to search has to be defined, as shown in Listing 5, following the concepts defined in Sect. 2. Then, an instance of the `Model_searcher()` functionality is created, defining

also the maximum number of trials to search for the optimum hyperparameters, and applying the random search methodology (Bergstra and Bengio (2012)). The model is then evaluated following a cross-validation scheme given by the number of folds into which the dataset is divided.

## 4 Application of NN4CAST to predict Northern Hemisphere SLP from the Pacific SSTs

In this section, a case study is performed to exemplify how the NN4CAST application could be used in a specific situation.

Concretely, in modelling the teleconnection between the El Niño-Southern Oscillation (ENSO) and the Northern Hemisphere (NH) climate variability in boreal autumn-winter, which is currently under debate (Yeh et al. (2018); Alizadeh (2024)). At seasonal time scales, ENSO is the globally dominant climate mode at interannual time scales. It corresponds to an air-sea coupled mode, characterized by anomalous warming or cooling that extends in a tongue-like shape from the central-eastern equatorial Pacific, coupled with a zonal gradient of SLP (Southern Oscillation) (Rasmusson and Carpenter (1982)). In this way, ENSO

generates a local atmospheric disturbance that propagates to remote regions as atmospheric wave trains, such as the Pacific North-American (PNA) or Pacific South-American (PSA) patterns (Straus and Shukla (2002)). Additionally, ENSO alters the



```
      from nn4cast.predefined_classes import Model_searcher

      params_selection =
          'pos_number_layers' = 5, # set the maximum value of fully connected layers (int).
 'pos_layer_sizes' = [16, 64, 256],# set the possible layer sizes (list).
          'pos_activations' = ["elu", "linear"],# set the possible activation functions
              (possibilities are all the ones availabe: tf.keras.layers.activations()) (list).
          'pos_dropout' = [0.0, 0.01],# set the possible dropout probabilities (list).
          'pos_kernel_regularizer' = ["l1_l2"],# set the possible kernel regularizer
 (possibilities are: l1_l2, l1, l2, None) (list).
          'search_skip_connections' = False,# set if searching for skip connections, either
              intermediate or end_to_end connections (bool).
          'pos_conv_layers' = 0,# set the maximum number of convolutional layers, the
              predictor field (X) must be 2D (int).
 'pos_learning_rate' = [1e-4,1e-3],# set the possible learning rates (list).

      outputs_bm_cross_validation = Model_searcher(dictionary_hyperparams=hyperparameters,
          dictionary_preprocess=dictionary_preprocess,
          dictionary_possibilities=params_selection, max_trials=10, n_cv_folds=8)
```

**Listing 5.** Python example to create an instance of `Model-searcher` function to search for the optimum hyperparameters under the possibilities given. The hyperparameter range was defined based on the most common values reported in specialized literature and the random search method (Bergstra and Bengio (2012)). Then, the best combination obtained is evaluated following the cross-validation scheme selected.

Walker circulation, potentially connecting with other tropical basins (Alexander et al. (2002)). ENSO teleconnections are robust in some regions, such as the tropical North Atlantic, but in other areas, like the North Atlantic and Euro-Mediterranean regions, they exhibit strong non-linear characteristics and are non-stationary over time. Factors leading to this behaviour may

be related not only to different ENSO spatial configuration or flavours (Eastern Niños (EP) and Central Niños (CP)) as well as ENSO intensity (King et al. (2023)), but also to potential decadal modulation patterns such as the Pacific Decadal Oscillation (PDO) and the Atlantic Multidecadal Variability (AMV) (López-Parages et al. (2015); López-Parages and Rodríguez-Fonseca (2012)).

In order to illustrate the application of the NN4CAST library, the model will be applied to predict the November-December

(ND) SLP anomalies over the whole NH with the information from Pacific SST anomalies from the previous season, i.e., September-October (SO). This choice was made for different reasons. First, the atmospheric response in the tropics, specially over the Pacific basin is more linear, expecting a good performance of the model over this region. Secondly, as seen in (López-Parages and Rodríguez-Fonseca (2012)), the teleconnection with the North Atlantic and Europe is non-stationary at all seasons, selecting this region to test the performance of the model in capturing this highly non-linear teleconnection. Additionally, the

election of ND season is in accordance with recent studies, which highlight a potential strengthening of the ENSO influence over this region on the early winter (Hou et al. (2023)) . The database chosen to construct the predictor is the HadISST (Rayner et al. (2003)), while the predictand field comes from ERA5 (Hersbach et al. (2020)). The regions selected for the predictor and



predictand are, respectively, the Pacific basin, ranging from $-20°$ S to $75°$ N and $120°$ E to $80°$ W, and the entire NH spanning from $-20°$ S to $75°$ N and $180°$ W to $180°$ E.

280 The parameters and hyperparameters used are the ones shown in Listing 1, so the same steps as defined in Listings 1, 2 and 3 are done. Specifically, first, the dictionary with the details of the simulation is created and saved in the outputs directory. Then, the preprocessing of the datasets is done by applying as optional main arguments: regriding the data to reduce the computational cost of the simulation and a detrend to remove the signal associated to the anthropogenic warming and low frequency variabilities by defining a sliding window of 15 years for both variables. Afterwards, the model is created following

285 the architecture defined in the dictionary of the hyperparameters in Listing 1. Then, the model is tested on the period 2000-2019 for comparing with the results of Hou et al. (2023), training it on 1940-1989 and validating on 1990-1999. The metrics of model performance on the test set are summarised in the output figure of the function (Fig. 2). In this case, the highest skill in terms of ACC (>0.7) between the predicted and observed ND SLP in the test period, is obtained at equatorial and tropical latitudes, specifically in the Pacific basin. This is in accordance with the higherair-sea coupling in this region and to the more linear

290 response of the atmosphere to the SSTs, which takes place at equatorial latitudes (Wang and Fiedler (2006)). Additionally, there is skill at extratropical latitudes, namely the North-Atlantic region. This is consistent with the findings of the study by (Hou et al. (2023)), which indicates a strengthening of the ENSO-East Atlantic pattern teleconnection from the 2000s onwards. Indeed, the centre of greatest skill in the model is located over the east of the British Isles (Fig. 2a). Remarkably, even better results could be achieved by training the model during a period when the ENSO teleconnection with this region is stronger, as

295 described by (Hou et al. (2023)), which is out of the scope of this article. However, there are regions with no or even negative skill, which may be due to the fact that there is no predictability from the Pacific SSTs over these regions in this test period, such as over southern Asia and Europe. When it comes to the time series of global ACC (Fig. 2b), it can be seen that it evolves in time, showing some variability between the first years (2000-2004) with reduced correlation on average, while the central years (2005-2010) are better predicted. Indeed, this figure could be used to analyze potential modulations of the teleconnection.

300 In this case, one could discuss the role played by other regions of SST like the Tropical North Atlantic (TNA) (An (2009)) or by other components of the Earth system. In terms of RMSE, the figure let us analyze possible errors in the amplitude of this variability. The highest errors are observed in regions with greater internal variability in SLP, such as along the storm track pathways of the Pacific and Atlantic, and at the highest latitudes where Pacific SSTs may not be the primary driver of SLP (Fig. 2c)

305 Nevertheless, to obtain a more robust measure of the model performance, it is convenient to apply a cross-validation technique. This is done by setting as True the cross-validation argument and defining the number of folds into which the dataset will be divided, as done in Listing 4. Specifically, a 4-fold is selected, meaning that the dataset is divided in 4 groups of 20 years, where the models are trained and tested iteratively (1940-1959; 1960-1979; 1980-1999; 2000-2019). Additionally, in order to analyze underlying physical dynamics, the attributions for the region comprised by [[$50°$ N,$65°$ N], [$25°$ W,$10°$ W]]

310 are calculated following the Integrated Gradients methodology detailed in Sec. 2. In this case, the metrics evaluating the performance of the model, depicted in Fig. 3, are slightly different than before. Concretely, the ACC map is smoother and less noisy globally (Fig. 3a). The skill in the tropical Pacific remains high, due to the high air-sea coupling over this region. Additionally,



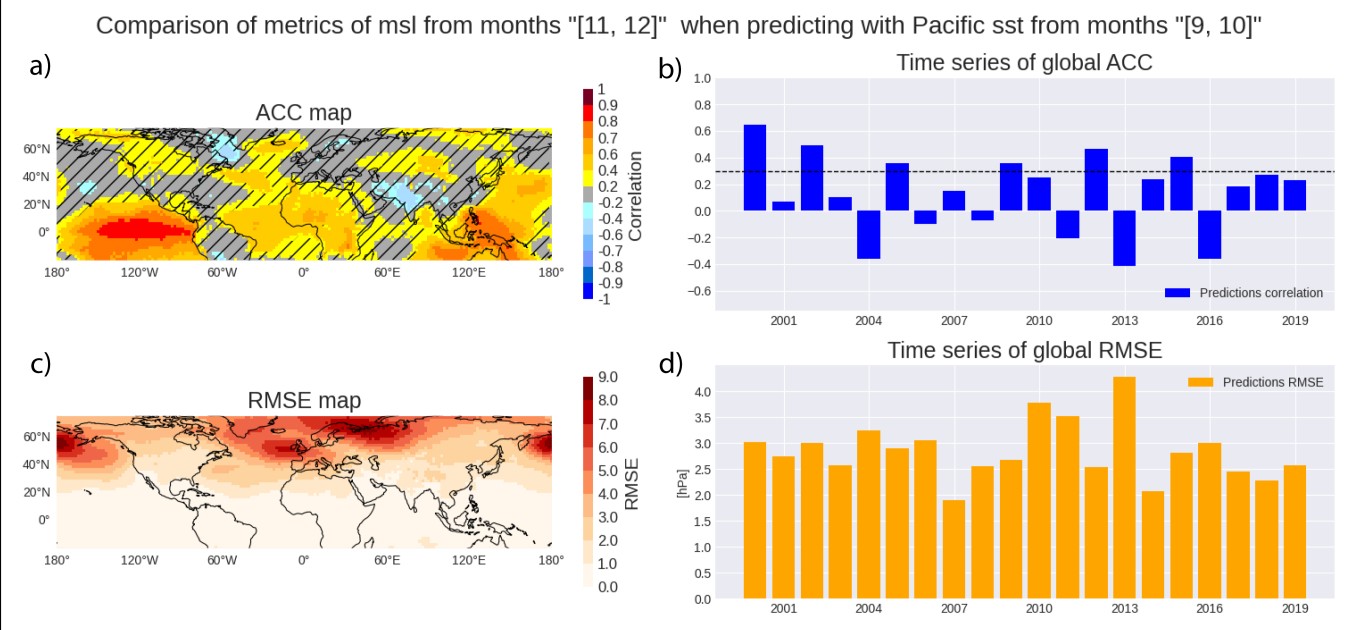

**Figure 2.** Panel of model performance metrics over the test period, when predicting the SLP anomaly field of ND using the Pacific SST anomalies from SO as the predictor, comparing it with the observed anomalies of ND SLP. Concretely in: a) ACC map; b) Time series of global ACC; c) RMSE map and d) Time series of global RMSE; all of them between predicted and observed fields. The ACC (RMSE) map indicates the correlation at each spatial point between the predicted and observed SLP anomaly time series during the period 2000-2019. While the time series of global ACC (RMSE) represents the correlation between the predicted and observed global mean SLP anomalies over time. The non-dashed regions plotted in figure a) represents the significant results according to a one-tailed T-test and the significance set in the hyperparameters of the simulation (in this case, 90% significance).

spurious negative correlations found in Fig. 2a over continental regions such as India, China or North-America are filtered. The decadal variability is more clear in the time series of global ACC (Fig. 3b), showing that the 1950s and the 2000s show

higher skill in terms of ACC while the 1990s are less predictable from the Pacific basin. In terms of RMSE, the results are similar than in the case before. In addition, to facilitate the analysis of the non-stationarity of the teleconnections, a panel of ACC maps is shown in which the model is tested on different periods according to those into which the model was split when the k-fold cross-validation was applied. Concretely, the dataset is divided in 4 folds, and for each iteration, the model is tested in two decades and trained in the rest, repeating this process iteratively. For instance, when testing the model on the 1940-1959

period, it is trained using data from 1960-2019. In this way, the model performance is evaluated across the different decades. This approach helps determine whether the predictability of North Atlantic SLP associated with Pacific SSTs varies in strength from one decade to another, allowing us to identify periods with stronger predictive skill (windows of opportunity) for seasonal forecasting. Figure 4 shows a comparison of model skill in different periods. After a quick inspection, it can be seen that the skill in terms of the ACC varies from one period to another, with better global results from the 1980s onwards. However, the



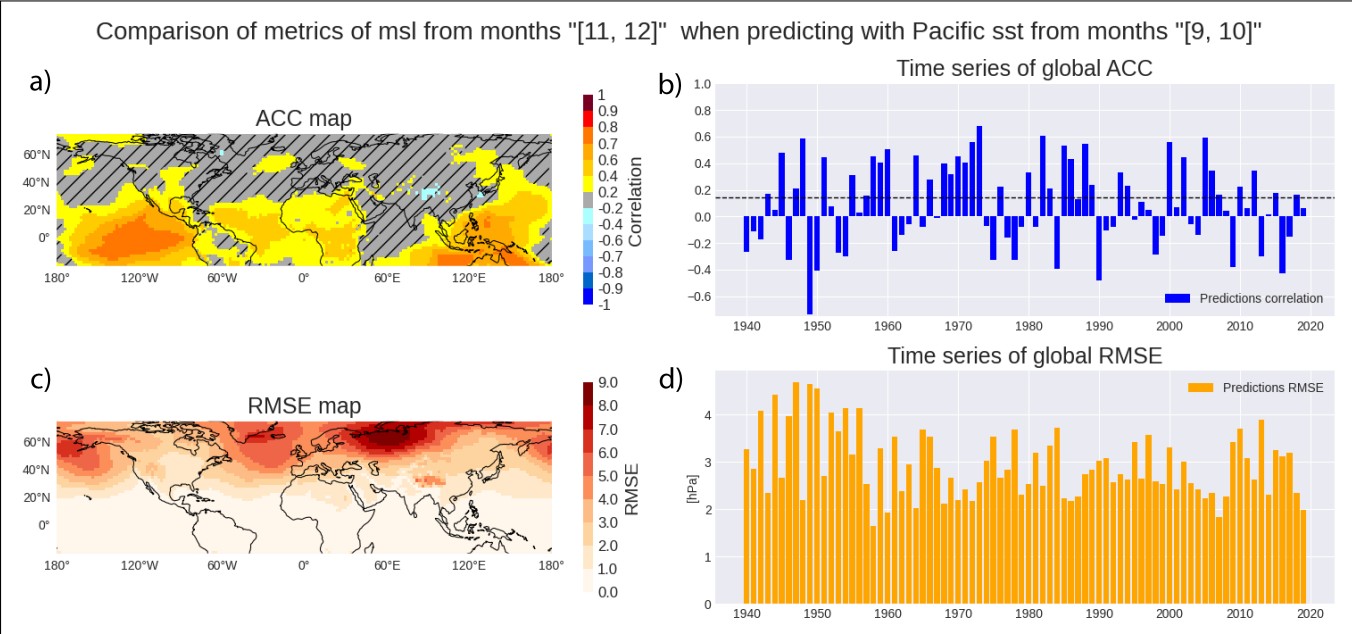

**Figure 3.** Panel of model performance metrics over the full period (1940-2019) by applying 4-fold cross-validation, when predicting the SLP anomaly field of ND using the Pacific SST from SO as the predictor, comparing it with the observed anomalies of ND SLP. Concretely in: a) ACC map; b) Time series of global ACC; c) RMSE map and d) Time series of global RMSE; all of them between predicted and observed fields. The ACC (RMSE) map indicates the correlation at each spatial point between the predicted and observed SLP anomaly time series during the period 1940-2019. While the time series of global ACC (RMSE) represents the correlation between the predicted and observed global mean SLP anomalies over time. The significant results of the simulation according to a one-tailed T-test and the significance threshold defined in the hyperparameters (in this case, 90% significance) is given by the non-dashed regions plotted in panel a) and the values above the dashed line of panel b).

skill in predicting equatorial latitudes remains consistently high (>0.4) across all time periods. In contrast, the skill in extratropical regions varies significantly, which aligns with the non-linear and non-stationary nature of these teleconnections. It is remarkable in this way the variability over the North-Atlantic. Concretely, in the decade of 1940-1959 (Fig. 4a), there is a global reduction of skill, specially in the extratropics. On the contrary, in the decades of 1980-1999 (Fig. 4c) and 2000-2019 (Fig. 4d), the skill is higher over this specific regions. It is remarkable the ability of the DL model to capture the behaviour of

this teleconnection over the different decades, taking into account that the model only have the information from the Pacific basin SSTs to make its predictions over the tropics and the whole Northern Hemisphere.

     Additionally, the `Results-plotter` function allows to get a visualisation of the model prediction and compare them with the observations for certain samples, concretely, for year 2015, represented in Fig. 5. On it, the predictor field is represented (Fig. 5a), showing a situation of a typical El Niño like signal, whereas, the prediction and observation for that year



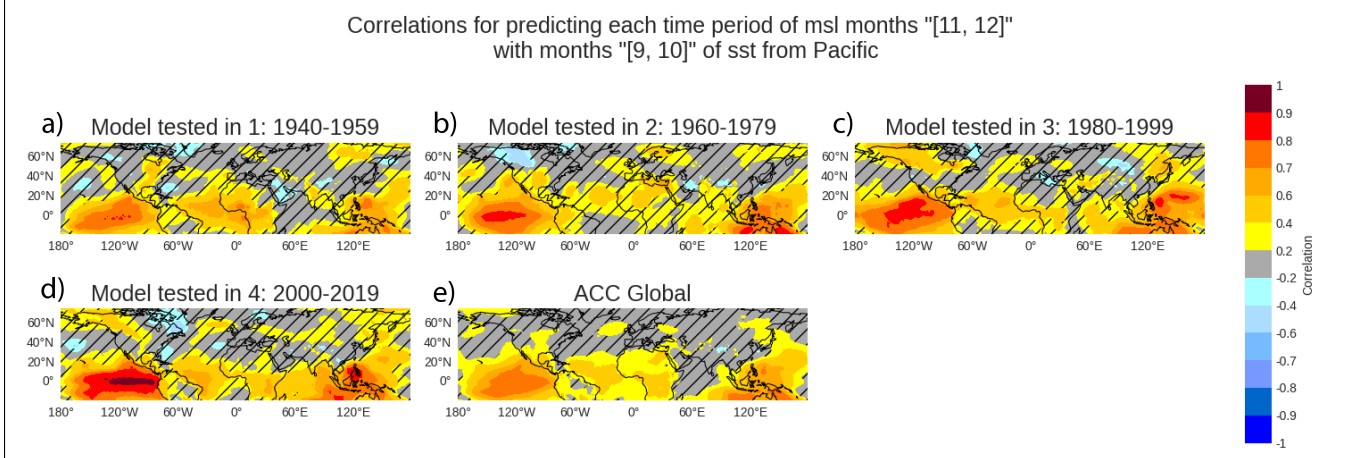

**Figure 4.** Panel of model performance in terms of spatial ACC over the 8-fold cross-validated periods. Concretely, each map depicts the spatial ACC between the predicted and observed ND SLP anomalies in the decades described, which is the one tested in each iteration during the cross-validation. The significant results of the simulation according to a T-test and the significance threshold defined in the hyperparamenters (in this case, 90% significance) is given by the non-dashed regions plotted in panel a) and the values above the dashed line of panel b

are represented with colours and isolines, respectively (Fig. 5b). It can be seen how the model correctly reproduces the signal coming from the equatorial Pacific to the extratropics, reaching the North-Atlantic.

Furthermore, to gain insight into the predictions, Fig. 6a illustrates the attributions of the predictors for the selected region (green box, Fig. 6b). The importance values are expressed in units of the predictand field (hPa), providing the individual contributions of each predictor (SSTs) to the SLP predictand over the region specified above the green box . It can be observed

that the eastern equatorial Pacific contributes to a reduction in the SLP anomaly in the target region, while the central part exerted a positive influence. Furthermore, the extratropical SSTs played a significant role in the prediction of this specific sample.

## 5  Conclusions

It is well known the importance of seasonal predictions for climate services. Earth System Models produce these predictions

by taking into account the interactions of the different components of the climate system. However, they are posed to different problems not only from the misrepresentation of some physical process due to the limited spatio-temporal resolution, but also due to errors in the initial and boundary conditions. Classical statistical approaches (linear regression, maximum covariance analysis, etc) focus on the relationships between a predictor and a predictand within a specified time lag between them, avoiding the problem of the numerical solving of the dynamical equations of the atmosphere. Nevertheless, their main drawbacks

rely on the limited number of past cases, the non-stationarity behaviour of certain relationships and the assumption of a



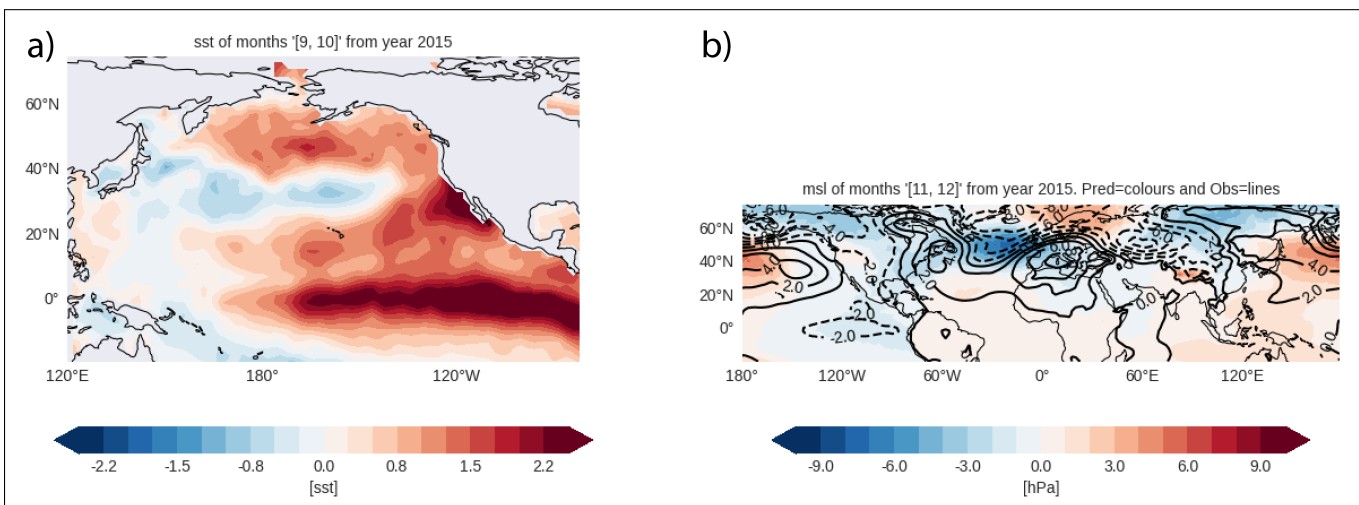

**Figure 5.** Left panel: Pacific SST SO (in ℃) predictor field for that specific year 2015. Right panel: comparison of model prediction (shaded in hPa) and observation (contours in hPa) of the SLP ND anomalies for year 2015.

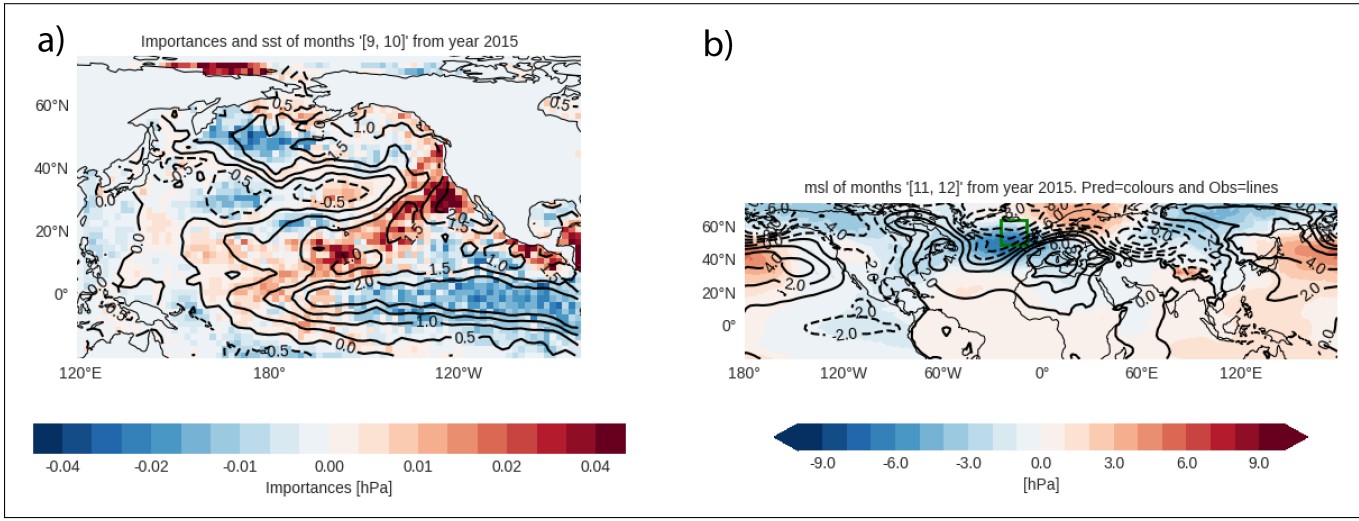

**Figure 6.** Left panel: Pacific SST SO predictor field for the specific sample (contours in ℃) and the attributions (shaded in hPa) for the predictand region selected (the green box in right panels) Right panel: comparison of model prediction (shaded in hPa) and observation (contours in hPa) of the SLP ND anomalies for year 2015.



linear relationship between the fields. It is for this reason that we have decided to develop a tool that allows modelling these relationships between fields from a non-linear perspective thanks to the application of deep learning, which is called NN4CAST. One of the main advantages of this modelling framework is its versatility and simplicity, as it does not require extensive knowledge of deep learning concepts and techniques by the user, making it easier for domain experts to apply deep learning

techniques in their respective fields of study.

The choice of the case study related to ENSO teleconnection predictability in early winter is motivated mainly because there is currently no consensus of how is the ENSO impact in this season over the extratropics, specially the North Atlantic European region. Recently, some studies has suggested a multidecadal variability in this teleconnection. Indeed, this is in line with the results shown in this paper, enabling the identification of potential windows of opportunity in certain periods and also allowing

the opportunity to deeper analyze the drivers of this predictability changes. This information is very useful for the scientific community, allowing also to improve and complement the actual dynamical models on seasonal predictions.

The NN4CAST framework not only allows for model assessment through metrics that identify such windows of opportunity, but also has a strong visual output display that provides the user to understand the character of the teleconnections that have been modelled, not only through metrics on model skill, but by enabling the user to analyse attributions in the predictive field

of predictions in an area of interest as well. This allows the identification of the most relevant areas for the model in each of the predictions, thus analysing possible physical mechanisms behind them. Additionally, this framework is designed to allow the integration of more complex deep learning architectures currently used in meteorological modelling, such as transformers or graph neural networks. This allows for the examination of the advantages and disadvantages of each architecture directly within the model.

In conclusion, the NN4CAST methodology enables the examination of the predictability of the climate system at subseasonal and seasonal time scales. Its main applications are as follows:

· The capacity to model non-linear relationships between fields, evaluating it through different skill and error metrics.

· Versatility and simplicity in its use by the user.

· The ability to detect windows of opportunity where the relationship between fields is more intense.

· The possibility of analysing the drivers of these changes in predictability through the model's attributions in the predictions.

· The potential for conducting sensitivity experiments.

*Code and data availability.* The current version of NN4CAST (1.0.19) is available from the Gihub repository at https://github.com/Victorgf00/nn4cast, under the MIT licence. The exact version of the model used to produce the results used in this paper is archived on Zenodo

Galván Fraile et al. (2024), as are input data and scripts to run the model and produce the plots for all the simulations presented in this paper.





*Author contributions.* The paper was written by VG, with contributions and suggestions from all the authors. The code was developed by VG.

*Competing interests.* The contact author has declared that none of the authors has any competing interests.

*Disclaimer.* Publisher's note: Copernicus Publications remains neutral with regard to jurisdictional claims made in the text, published maps,
institutional affiliations, or any other geographical representation in this paper. While Copernicus Publications makes every effort to include appropriate place names, the final responsibility lies with the authors.

*Financial support.* This research has been supported by the Spanish Ministry of Science, Innovation and Universities through the National Program FPU (grant no. AP-2022-02162) and the Oceans for Future project (Innovative climate services using ocean information and communication with society. grant no. TED2021-130106B-I00 funded by MCIN/AEI /10.13039/501100011033 and by the European Union
Next GenerationEU/ PRTR Strategic Projects oriented to the Ecological Transition and the Digital Transition. Call 2021). MMR has been supported by Ramón y Cajal (RYC2022-038454-I, funded by MCIN/AEI/10.13039/501100011033 and co-funded by the FSE+,European Union)



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
