# Peer review of "NN4CAST: An end-to-end deep learning application for seasonal climate forecasts"

_EGUsphere, 2024_

## Author Comment (AC2)

**Response to Reviewer**

November 22, 2024

**Detailed Comments**

Responses are marked in blue.

Anonymous Referee #1, 24 Oct 2024:

*This is an interesting application for building sub-seasonal models but I have several concerns that would be good to address before it is published.*

Dear Referee,

Thank you very much for your positive feedback and for the time and effort you dedicated to reviewing our manuscript and dataset.

We greatly appreciate your insightful comments, which have been invaluable in guiding improvements to our manuscript. Please find below a detailed, point-by-point response outlining our approach to addressing each of your suggestions. If you feel any of these adjustments might not fully meet the needs you highlighted, we would be grateful for further guidance.

Kind regards,
Víctor Galván (on behalf of the author team)

*Major Corrections*

*1. It is not clear to me what the focus of this paper. Is it to present this new framework and the model you have trained is just an example of an application that could be done with the new framework? Or is the idea to present this new model? If the former, do you have plans to extend this into short lead time weather forecasting? It seems to me that most of what you have developed e.g. hyperparameter tuning and XAI could be useful here. If the latter then I think it would be good to have a better description of the model.*

Thank you for your valuable comments on the focus of the paper. The primary objective of the paper is to present a framework that enables users to develop deep learning models with a specific focus on sub-seasonal and seasonal forecasting. The principal benefit of this framework is that it enables the user to undertake preprocessing, training and validation of the model in a straightforward manner. To exemplify this, we have selected a case study in which the Pacific sea surface temperature anomalies in September and October are employed as the predictor field, while the global sea level pressure anomalies in November and December constitute the predictand field. At the outset, the intention was not to extend this model to encompass short-lead-time weather forecasting. Nevertheless, if required by the user, an updated version of the model could

be developed to enable this, thereby facilitating a comparison of its performance with that of other existing dynamic and machine learning models. However, this is out of the scope of the present study.

In this new version, we have highlighted the main goal of the paper and emphasised the flexibility of the tool presented to develop deep learning models (Lines 83-88):

*"For these reasons, we developed the Neural Network foreCAST (NN4CAST) application, a Python library designed to facilitate the creation of deep learning models for non-linear modelling of climate teleconnections. One of the main objectives of NN4CAST is to avoid treating these deep learning models as "black boxes", enabling users to analyse the origins of the predictability and assess the sensitivity of predictions to changes in the training period."*

*2. As you mention in the introduction, sub-seasonal forecasting is very uncertain. I think for this framework to have a significant impact, it would need to be able to include a way to quantify uncertainty. For example, allowing multiple initial conditions, injections of Gaussian noise or the generation of ensembles.*

We would like to thank you for this insightful comment. We are in full agreement with the proposal to provide a measure of the uncertainty of the model predictions. This might take into account both potential errors in the modelling of the teleconnections and the ability to analyse non-stationary behaviours of the teleconnections. In terms of the methodology for quantifying this, a new function (based on the existing "Model_build_and_test" one) has been created that allows the generation of ensembles by training the model in different periods of the training set. This enables the sensitivity of the model to these periods in the different regions to be evaluated. Specifically, what this new function does is to create a given number of models that differ only because they have been trained on different datasets, using the bagging (bootstrap aggregation) method. In this way, it quantifies the uncertainty associated with the period chosen to train the model and how this depends not only on the region we are analysing, but also on the teleconnection mechanisms involved. However, this function has not been implemented in the library, as it can be easily created and tailored by users to suit their specific needs, optimising it for the particular region and teleconnection mechanisms under analysis.

Furthermore, another method for introducing uncertainty into the model is to vary the initialisation seed of the trainable parameters of the model, which is already incorporated into the application. Nevertheless, our analysis of the case study revealed that variations in the seed do not result in significant alterations to the model predictions. Nevertheless, in other scenarios, this could prove to be a more pertinent consideration, as evidenced in the article of Scher, S., & Messori, G. (2021), where they tested different ways to generate ensembles, and how this could lead to better overall performances and

uncertainty estimates. Furthermore, as you have stated, uncertainty can be estimated by utilising multiple initial conditions or by incorporating noise into them. This issue has been addressed by allowing the user to modify the predictor field files before introducing them as inputs to the model, despite the absence of a predefined function for this purpose within the application. Basically, the user just needs to add some noise to the data files (predictor and/or predictand) before doing the preprocessing phase.

We have highlighted in the discussion the possibilities of the model to perform sensitivity experiments by not only using different predictor and predictand fields, but changing the regions, the datasets and introducing noise to them to quantify the uncertainty (Lines 362-366):

*"The NN4CAST framework also supports the development of sensitivity experiments, allowing users to explore not only different predictors and predictands, but also variations in the regions of them, different datasets and the introduction of noise. For instance, an attempt was made to introduce white noise into the predictor during the test period. However, this resulted in only minor alterations, indicating that the trained model is resilient to this type of noise in this specific case. These capabilities significantly enhance its versatility for exploring and understanding climate predictability."*

*3. It is unclear to me who provides the model? Are there example models provided in the repository? Or can the user prepare their own models and what format should they be in? Torch/tensorflow?*

We are grateful for your observation regarding the construction of the model. The principal benefit of this framework, which to our knowledge is unique at least at seasonal timescales, is that the model code does not need to be programmed; it is fully implemented. The user is required to select the predictor and predictand, provide the initial hyperparameters (number of convolutional layers, activation functions, etc.) and then, the model will be created, trained and validated based on this information.

This application is founded upon the utilisation of pre-existing libraries, including TensorFlow and NumPy. Nevertheless, any neural network-based model constructed with alternative libraries, such as PyTorch, can be employed within the application. In this instance, the user would be required to program the model and utilise the library preprocessing and validation functions to assess the overall performance of their model.

*4. For the model you present, I am not convinced that cross-validation is appropriate across an annual timescale. Is the idea to make the model robust against climate change? We know that ERA5 is also worse pre-1979 because of the lack of satellite observations.*

We highly appreciate your contribution to the discussion. In this instance, detrending was conducted using the backward moving average method (using a sliding window of 50 years) in both fields with the objective of evaluating the predictive capacity of the model in terms of the internal variability of the climate system. The quality of data from the pre-satellite era, while subject to limitations, including reduced confidence in assimilated observations within ERA5, still allows for the detection of low-frequency signals. This is evident from the consistent oscillations observed when training with periods both before and after the 1970s.

In addition, and taking the above clarifications into account, the model allows the user to test the model for different periods changing the number of folds ,and not just a leave-one-out cross-validation. Figure 4 in the text (attached below) allows for a comparison of the model skill in terms of the anomaly correlation coefficient (ACC) between the predictions and the observations in the different folds of the cross-validation. To illustrate, in fold 4, where the model was trained on data from 1940 to 1999 and tested on data from 2000 to 2019, it exhibited a significant positive correlation not only in the tropics but also in the extratropics. This is despite the fact that the model was trained on data from a period with potentially inferior data quality. It may therefore be concluded that despite the potential errors in the data, the model is capable of learning the underlying mechanisms from these signals and extrapolating them to new unseen cases during training.

[Figure]

**Figure 4.** Panel of model performance in terms of spatial ACC over the 8-fold cross-validated periods. Concretely, each map depicts the spatial ACC between the predicted and observed ND SLP anomalies in the decades described, which is the one tested in each iteration during the cross-validation. The significant results of the simulation according to a T-test and the significance threshold defined in the hyperparameters (in this case, 90% significance) is given by the non-dashed regions plotted in panel a) and the values above the dashed line of panel b

*Minor*

*1. I think in the discussion around line 65, it would be good to mention Neural GCM as an example of an effective hybrid model.*

We are pleased to receive this appreciation. We agree that this new model (Neural GCM) serves as an excellent illustration of how a hybrid deep learning model, trained to make short-term weather predictions, is capable of making forecasts on seasonal and

decadal timescales for different atmospheric variables. Furthermore, it serves to elucidate some challenges of running this type of hybrid models at longer timescales (numerical instabilities and climate drifts). For this reason, in this new version, we have added this example in line 67:

*"Some examples of these data-driven models are: PanguWeather, which is a 3D Earth-specific transformer module created by the Huawei Cloud group (Bi et al. (2022)); GraphCast and NeuralGCM, developed by researchers from DeepMind and Google (Lam et al. (2022); Kochkov et al. (2024))."*

*2. I am not sure Lines 73-74 follow. You say the models are largely linear and then you say that this is important for non-linear relationships?*

We will try to provide a more detailed clarification of this sentence in the text. However, the underlying concept is that classical statistical models have the limitation of assuming a linear relationship between predictor and predictand variables (Wilks, D. S. (2011)), which is a significant drawback when modelling complex systems such as the climate system, where the processes' interactions are highly non-linear. In this new version, we have clarified this topic in Lines 73-75:

*"The underlying assumption of linear relationships between predictor and predictand fields is a common premise in these models. However, this assumption may not be entirely appropriate when modelling the Earth system, which is mainly composed of complex non-linear components."*

*3. I think large parts of Section 2 could be removed. The basic theory of neural networks does not need to be included in a paper.*

Thank you for your comments. The principal aim of this application is to provide a tool that enables climate experts to utilise non-linear modelling through the application of deep learning techniques, without the necessity to programme these models from their fundamental principles and without the requirement of a comprehensive understanding of this field. This section is designed to provide an overview of the fundamental principles involved, with the aim of ensuring that all users of this application are able to understand the various concepts associated with such techniques. However, in light of the comments received, some of the basic theory has been reduced in length and the reader is directed to literature on the subject for further information.

---

## Author Comment (AC3)

**Response to Reviewer**

March 04, 2025

**Detailed Comments**

Responses are marked in blue.

Review of NN4CAST: An end-to-end deep learning application for seasonal climate forecasts, submitted to GMD, February 2025.

Brian Henn, Allen Institute for Artificial Intelligence, climate modeling team, Seattle, WA

Dear Referee,

Thank you very much for your valuable feedback and for the careful review of our manuscript. We truly appreciate the time and effort you have dedicated to this process.Your insightful suggestions have been extremely helpful in enhancing our work. Please find below our detailed responses to each of your comments. If there are any aspects that you feel require further refinement, we would be grateful for your additional input.

Kind regards,

Víctor Galván (on behalf of the author team)

General comments

*The authors describe a software package ("NN4CAST") that allows for training neural networks (NNs) to predict climate model fields in a simple, single-input to single-output manner. The software also allows for some plotting and interpretability of the trained NN model and its predictions. The authors describe what neural networks are, and how they may fit into seasonal predictions of climate variables, at a fairly elementary level. They then present an example in which their package has been used to predict monthly-averaged SLP fields from monthly-averaged SST fields with a lead time of about two months, showing where the predictions have more and less skill and providing some examples of the ML interpretability and attribution.*

*NN4CAST could provide a nice educational example of using simple NNs to predict single fields as outputs of other single fields. This type of software would help students learn about basic ML and NN implementations, and allow them to understand when simple predictability can be leveraged via these tools. It is nicely implemented from a usability standpoint, with flexible configurations for data preprocessing and simple NN*

*model training/hyperparameter tuning. But, for the reasons I note below, it is unlikely to provide state of the art predictability improvements in the field of seasonal forecasting.*

*The authors note that seasonal forecasting may be nonlinear and that conventional statistical techniques are primarily linear, as a justification for using neural networks for season forecasting. However, they state that they choose an example (Northern hemisphere wintertime ENSO response) that they state is mostly linear. Thus, it is likely that their results would be the same if they used an off-the-shelf linear approach (eg linear regression) that does not require NN training. In my opinion, there would be much stronger justification for the package if the authors could show that it produces improved results in a nonlinear case, beyond what a linear baseline approach could achieve.*

Thank you for your comment. We would like to apologize for the misunderstanding. We selected the Northern Hemisphere early winter ENSO response as it allows us to analyze how our model performs in different regions. On one hand, in the tropics, where the response to ENSO is predominantly linear, and on the other hand, in the extratropics, where nonlinear components play a fundamental role. This setup enables us to assess the behaviour of our model in both contexts. To overcome this misunderstanding in the text, we have added updated Lines 73-75:

*"The underlying assumption of linear relationships between predictor and predictand fields is a common premise in these models. However, this assumption may not be entirely appropriate when modelling the Earth system, which is mainly composed of complex non-linear components."*

To further illustrate this point, we have implemented a baseline model based on a multiple linear regression approach, trained in the same manner as the neural network for this specific case. Specifically, both models were trained using the same preprocessed data from the NN4CAST library, following a four-fold cross-validation scheme, as described in the manuscript. As evaluation metrics, we compared both the Anomaly Correlation Coefficient (ACC) and the Root Mean Square Error (RMSE).

In figure A1, we present the results obtained using the linear regression (first row) and the outcomes for the neural network (second row). The third row displays the differences between the two approaches. It is important to note that both methods aim to minimize the Mean Squared Error (MSE) in this case, meaning that the primary improvements expected from the neural network should be reflected in the RMSE. This is precisely what we observe, particularly in the extratropical regions, where the magnitude of the anomalies and, thus, the errors are higher.

[Figure]

**Figure A1.** Panel of models performance over the full period (1940-2019) by applying 4-fold cross-validation, when predicting the SLP anomaly field of ND using the Pacific SST from SO as the predictor, comparing it with the observed anomalies of ND SLP. Concretely in: a) and b) ACC and RMSE maps for the multiple linear regression model, respectively; c) and d) ACC and RMSE maps for the neural based model, respectively; e) and f) ACC and RMSE differences between the two models. In e) the blue colors represent a better performance of the NN model, while red colors represent better performance of the multiple linear regression model. In f), red colors represent less error by the NN model, while turquoise colors less error by the multiple linear regression

We have added some clarification in the text about this around Line 290 :

*"In this case, the highest skill in terms of ACC (>0.7) between the predicted and observed ND SLP during the test period is obtained at equatorial and tropical latitudes, specifically in the Pacific basin. This is in accordance with the higher air-sea coupling in this region and to the more linear response of the atmosphere to the SSTs, which takes place at equatorial latitudes (Wang et al. (2006)). Additionally, there is skill at extratropical latitudes, particularly in the North-Atlantic region, which is characterized by the dominance of highly non-linear signal interactions (Hurrell et al. (2010)). This is consistent with the findings of the study by (Hou et al. (2023)), which indicates a strengthening of the ENSO-East Atlantic pattern teleconnection from the 2000s onwards. Indeed, the centre of greatest skill in the model is located over the east of the British Isles (Fig. 2a). We have explicitly examined the differences between our model and a multiple linear regression approximation, observing improvements in both tropical and extratropical regions (not shown). Remarkably, even better results could be achieved by training the model during a period when the ENSO teleconnection with this region is stronger, as described by (Hou et al. (2023)), which is out of the scope of this article."*

*Relatedly, most of the benefits of using NNs come from "deep" learning where datasets are large and have complex relationships between predictors and predicands. NN4CAST as applied here is not capable of tackling these kinds of problems: for models with many deep layers and large numbers of parameters because training will become impractically slow without GPU support. As such, NN4CAST as currently written is limited to "toy" examples, where it perhaps does not outperform linear regression. The software is also written in simplistic ways that will not allow it to be extended to other datasets and large-scale use cases. For example, it does not use typing, object-oriented organization, etc. that would make it easier to modify to work on other cases.*

Thank you for your comment. Indeed, we should have emphasized this aspect more clearly in the manuscript to better differentiate our model from other operational deep learning models that rely on substantial GPU support. NN4CAST is a deep learning tool specifically designed for conducting sensitivity experiments efficiently in seasonal forecasting studies. As currently implemented, it allows for the evaluation of the predictability of a field from a predictor (in our case, sea surface temperatures), facilitating not only an assessment of model skill but also enabling modifications to various parameters such as the forcing region, seasons, forecast times, lead times, and training windows.

The strength of NN4CAST lies in its ability to deliver reliable results with lower spatial resolution and using a single predictor and predictand field within the observational record, all while keeping computational costs to a minimum. Additionally, due to the relatively small datasets utilized, there is no need for parallelization, making the model more computationally efficient. While the software is not designed for large-scale datasets, it is written in an object-oriented manner, and its clear and simple structure ensures that it could be adapted to other data-handling environments or more efficient computational frameworks.

We acknowledge the potential for future improvements and, in line with your suggestion, are open to incorporating features such as pretraining on larger datasets (e.g., historical CMIP simulations) and applying transfer learning to refine the model with observational data. However, while the primary goal of this model is not to produce operational seasonal forecasts, its intended purpose extends beyond serving as a mere teaching tool. Instead, it provides a complementary approach to traditional models, offering insights into the expected impact of a given SST state and enabling comparisons with dynamical model results. This can enhance the interpretation of forecasts.

For instance, we have generated a forecast for the 2024 ND SLP anomalies using September SST data and compared them with predictions from the ECMWF SEAS5 model (October initialization) as well as with ERA5 reanalysis data. As illustrated in Figure A2, the NN-based model produced a more accurate extratropical response than the dynamical model, better capturing the extratropical anomaly pattern.

[Figure]

**Figure A2.** Panel showing ND 2024 SLP anomalies over the Northern Hemisphere. Specifically: (a) ERA5 reanalysis anomalies, (b) NN4CAST-predicted anomalies, and (c) SEAS5 (ECMWF)-predicted anomalies. The climatological reference period is 1981–2016. The correlation and RMSE between the predicted and observed anomalies for both models are indicated in brackets.

*Overall, however, the paper is clearly written, the figures are mostly quite informative, and the examples of software code snippets are quite helpful. As I mentioned, this is a useful tool for teaching beginner users of ML how to apply it towards earth science data.*

Thank you for your comment. However, we do not fully agree with the characterization of the model's utility. While it is true that the primary goal of this model is not to produce operational seasonal forecasts, its purpose extends beyond serving solely as a teaching tool. Rather, it is intended as a complementary approach to traditional models.

Specifically, it can be used to analyze the expected impact of a given predictor state and compare the results with those from dynamical models. This provides additional insights that may help improve forecast interpretation. In this regard, we have generated forecasts for ND 2024 using September SST data and compared them with predictions from the ECMWF SEAS5 model, as well as with ERA5 reanalysis data. As shown in Figure A2, the NN-based model provided a more accurate extratropical response than the dynamical model, better reproducing the extratropical anomaly pattern.

*Specific comments*

*L35: "Multidecadal ocean variability and the Global Warming trend alters the global circulation and, thus, the way in which atmospheric teleconnections (i.e., Rossby waves)*

*propagate, introducing non-stationarities in the system". It is probably worth noting that non-stationarity likely is a bigger problem for statistical models than it is for dynamical ones/ESMs, which can generalize to unseen regimes via physical laws. As a result this is not a very good justification for statistical approaches like NNs, unlike the other items in this list which do suggest that statistical approaches will be useful.*

You are correct that dynamical models inherently have the ability to account for such modulations. In this case, our intention was not to suggest that non-stationarity offers an advantage for statistical models, but rather to emphasize that the way these methods are trained can result in significantly different outcomes, depending on the training period, thus allowing for the detection of non-stationarities in the relationships. We have clarified this in the text:

*"Multidecadal ocean variability and the Global Warming trend alter the global circulation and, thus, the way in which atmospheric teleconnections (i.e., Rossby waves) propagate, introducing non-stationarities in the system (Lopez et al. (2015); Weisheimer et al. (2017)). While ESMs can generally extrapolate well to new regimes due to their reliance on physical laws, statistical models are more dependent on the training period and may struggle to adapt to changing relationships."*

This is illustrated in Figure 4 of the paper, where we observe that the validation results vary depending on the temporal window used. These differences are particularly pronounced in regions where decadal modulation is stronger. This could suggest that an optimal approach may be to train the models during periods when the relationship between the variables remains more stable.

[Figure]

**Figure 4.** Panel of model performance in terms of spatial ACC over the 4-fold cross-validated periods. Concretely, each map depicts the spatial ACC between the predicted and observed ND SLP anomalies in the decades described, which is the one tested in each iteration during the cross-validation. The significant results of the simulation according to a T-test and the significance threshold defined in the hyperparameters (in this case, 90% significance) is given by the non-dashed regions plotted the panels.

*L61-L74: The authors note growth of ML weather models here, but should be clearer that these types of models are making autoregressive forecasts of the evolution of the*

*atmosphere (and in some cases the coupled earth-ocean system, eg, this paper: https://arxiv.org/abs/2409.16247) at sub-daily temporal frequency. Some of these ML models are capable of making seasonal forecasts in this way. This is a much different and harder problem than making predictions of static snapshots of monthly-mean variables, which do not capture the high-frequency temporal variability of the climate. In my opinion the authors need to make the distinction clearer between Graphcast, etc., and NN4CAST in that regard – they are not really comparable in terms of what they forecast.*

Thank you for your comment. We have revised this section of the text to clarify the differences you mentioned. We have added this clarification around Lines 80-85. We attach here the new paragraph:

*"For these reasons, we developed the Neural Network foreCAST (NN4CAST) application, a Python library designed to facilitate the creation of deep learning models for modelling of climate teleconnections, including non-linear relationships. It is important to highlight the differences between our approach and other machine learning-based weather models. In models such as GraphCast or Pangu, predictions are made autoregressively, meaning they generate sub-daily forecasts of the Earth system's state while accounting for its temporal evolution. In contrast, NN4CAST models the relationship between static, monthly-mean variables. This approach allows for the development of a simpler model, reducing the risk of treating deep learning methods as ``black boxes'' and enabling users to analyze the sources of predictability and assess the sensitivity of predictions to variations in the training period."*

*L102: "2 Theoretical framework": While the material in this section is useful for beginners to the field, it is mostly what would be covered in an elementary textbook on machine learning. For that reason, it is likely not necessary to cover in this article. The authors can assume that readers either know this information already or can read it in other sources.*

Thank you for your comment. In response to feedback from another referee, we have significantly shortened this section. However, we would like to emphasize that the primary goal of this tool is to enable climate experts to apply model climate teleconnections using deep learning techniques without the need to program these models from the ground up or possess an in-depth understanding of machine learning principles.

This section provides an overview of the basic concepts involved, ensuring that users can grasp the key ideas behind these techniques. Nevertheless, in light of your comment, we have streamlined some of the introductory theory and have directed readers to relevant literature for further details.

*Equation 3: This is not the formula for ACC. This is a formula for a correlation coefficient between the predictions and the targets. ACC, however, is more complex: It is the correlation between anomalies from the mean prediction vs. anomalies from the mean target. See for example: https://wattclarity.com.au/other-resources/glossary/other-resources-glossary-anomaly-correlation-coefficient/*

Thank you for the clarification. We have explicitly stated in the equation 3 that, in this case, both the observations and predictions represent anomalies. We attach here the new paragraph:

*"Once the model has been trained, the parameters are fixed and used to make the predictions over the testing dataset. To evaluate its performance, different metrics could be used, such as the Root Mean Squared Error (RMSE), which is the square root of the MSE, or the Anomaly Correlation Coefficient (ACC), which is given by:*

$$ACC = \frac{\sum_{i=1}^{m} y^{(i)} \cdot \hat{\mathbf{y}}^{(i)}}{\sqrt{\sum_{i=1}^{m} \left( y^{(i)} \cdot \hat{\mathbf{y}}^{(i)} \right)^2}}$$

*where $\mathbf{\hat{y}}^{(i)}$ is the predicted anomaly of the predictand variable for the $i^{th}$ instance in the dataset and $y^{(i)}$ is its corresponding observed anomaly."*

*Listing 3: There is a typo here - the output of the first line is "outputs_hold_out", but the inputs to the second line include "outputs_cross_validation" that is not defined in this block.*

Thank you very much for pointing that out! You are correct, and we have made the necessary corrections. The output of the first line should indeed be "outputs_hold_out," and we have ensured that the second line now properly references the correct variable.

*Figure 2: Note that it is more typical to use stippling for areas where there \*\*is\*\* significant correlation, rather than areas where there is \*\*not\*\* significant correlation as Figures 2, 3, and 4 do. I would suggest using the more conventional approach.*

Thank you for your suggestion. However, we believe that adding stippling to areas where the results are not significant makes it easier to quickly discard those regions and focus on the meaningful signals. This approach enhances clarity and facilitates interpretation. Additionally, institutions such as the Barcelona Supercomputing Center and the Intergovernmental Panel on Climate Change (IPCC) also follow this convention in their reports.

*L310: "In this case, the metrics evaluating the performance of the model, depicted in Fig. 3, are slightly different than before. Concretely, the ACC map is smoother and less noisy globally (Fig. 3a)." I think this is misleading. My guess is that the reason the correlation maps are less noisy is not due to the k-folds approach, but simply that the map in Figure 3a is computed over a 4x longer period than the map in Figure 2a. A more correct comparison of the effects of k-folds would be to use the same test period (2000-2019) both with and without k-folds.*

Thank you for your comment. As you correctly point out, the smoothing of the ACC map in Figure 3a is indeed due to the model being validated over a longer period (using k-fold cross-validation), whereas in Figure 2a, it is only tested over the 2000–2019 period. To allow for a clearer comparison, we agree that it is more appropriate to evaluate both cases over the same period. For this reason, in Figure 4d, we present the ACC of the model specifically for the 2000–2019 fold.

However, it is important to note that there are slightly differences between both maps. In Figure 2a, the model is trained on the 1940–1989 period and validated on 1990–1999. Here, training stops when the validation error does not decrease over several consecutive epochs. In contrast, in the cross-validated model, the validation set is chosen as 10% of the data in each fold, leading to a different validation period and, consequently, slightly different results. Therefore, while the longer validation period does contribute to the smoother appearance, the primary reason for using k-fold cross-validation is to ensure that the model is evaluated across the entire available period rather than being limited to a single testing set.

We have added this clarification in the original text around lines 310-312:

*"In this case, the metrics evaluating the performance of the model, depicted in Fig. 3, are slightly different than before. Specifically, the ACC map appears smoother and less noisy globally (Fig. 3a). This difference arises not only due to the use of k-fold cross-validation, but also because the map in Fig. 3a is computed over a longer period compared to the map in Fig 2a , which is based on the shorter 2000-2019 testing period."*

*Figure 5b - It is visually difficult for me to assess the skill of the model in 5b when the predictions are plotted in shading and the observations are plotted in contours. An easier way for reasons to assess the skill of the predictions would be to plot two maps (one of shaded predictions and one of shaded observations, or one of shaded predictions and one of shaded model error with the same colorbar) side-by-side.*

Thank you very much for the suggestion. We have redesigned the figure accordingly and have attached the updated version below:

[Figure]

**Figure 5.** Comparison of model prediction a specific sample. Concretely: a) Pacific SST SO (in °C) predictor field for year 2015; b) model predictions and c) observed SLP ND anomalies (in hPa) for year 2015. The yellow box plotted in panel b) represents the region for which the attributions of the model are computed.

*Figure 6b: This seems to just repeat Figure 5b -- does it need to be shown again?*

Thank you for your insightful observation. You are correct that Figure 6b was redundant. We have redesigned the figure to enhance clarity and eliminate repetition. Accordingly, we have changed in the version of the manuscript this figure. Please find the revised version attached below.

[Figure]

**Figure 6.** Pacific SST SO predictor field for the specific sample of year 2015 (contours in °C) and the attributions (shaded in hPa) for the predictand region selected (yellow box in Fig. 5b).

*Technical corrections*

*Note: I attempted to download the zipfile of the datasets used from Zenodo, and found that I was not able to uncompress the zipfile (an error occurred). I am not sure if this data is correctly archived.*

Thank you very much for bringing this to our attention. We had not noticed this issue before, but we have now updated the file on Zenodo. You can access the corrected dataset using the following link: https://doi.org/10.5281/zenodo.14918750.

---

## Author Comment (AC4)

**Response to Reviewer**

March 04, 2025

**Detailed Comments**

Responses are marked in blue.

Anonymous Referee #1, 4 Dec 2024:
*Dear authors,*

*Thank you for your responses. Please find below my responses:*

Dear referee,

Thank you once again for your thorough review and valuable feedback. We appreciate the time and effort you have dedicated to evaluating our work. Please find our responses to your latest comments below.

*1. I'm afraid I still fail to see why this framework is specific to S2S. I think it would actually make a stronger paper if you said it could be used for doing short range forecasting too.*

Thank you again for your valuable feedback. Our original intention was not to extend NN4CAST to short-range forecasting, as this would involve adapting the model to handle the larger volume of available data and the increased computational costs associated with such applications. In particular, the complexity of short-range forecasting models would likely require the use of GPUs and parallelization to ensure efficiency and scalability. However, for future extensions of the framework, we are open to exploring the feasibility of adapting NN4CAST for short-range forecasting. This could involve modifications to the model, such as optimizing computational processes, to accommodate the demands of shorter lead times while maintaining the framework's usability and interpretability.

*3. If this is the case then I think there needs to be more details on the model used. By this I do not mean the neural network theory (of which I still think there is too much) but instead the architecture you used for the results you showed e.g. number of layers etc.*

Thank you once again for your valuable feedback. In line with our previous response, the details of the model architecture used for the presented results are explicitly provided within the listings that include the example code. These listings not only outline the preprocessing steps but also specify the fundamental hyperparameters necessary for the model's construction, such as the number of layers, activation functions, and other key settings. This ensures full transparency regarding the model configuration and provides users with a clear framework to modify and adapt the architecture to their specific requirements if needed.

*4. Please could you add some of this clarification to the paper to make it more understandable for the reader.*

Thank you for your suggestion. We have incorporated these clarifications into the manuscript to better explain the rationale behind our approach, including the use of cross-validation across different time periods and the considerations regarding data quality in the pre-satellite era.